# Climate-induced decline in the quality and quantity of European hops calls for immediate adaptation measures

Martin Mozny [1,2,3] ✉, Miroslav Trnka [1], Vojtech Vlach [3,4], Zdenek Zalud[1], Tomas Cejka[1], Lenka Hajkova[3], Vera Potopova [1,2], Mikhail A. Semenov [5], Daniela Semeradova[1] & Ulf Büntgen[1,6,7,8]

A recent rise in the global brewery sector has increased the demand for high-quality, late summer hops. The effects of ongoing and predicted climate change on the yield and aroma of hops, however, remain largely unknown. Here, we combine meteorological measurements and model projections to assess the climate sensitivity of the yield, alpha content and cone development of European hops between 1970 and 2050 CE, when temperature increases by 1.4 °C and precipitation decreases by 24 mm. Accounting for almost 90% of all hop-growing regions, our results from Germany, the Czech Republic and Slovenia show that hop ripening started approximately 20 days earlier, production declined by almost 0.2 t/ha/year, and the alpha content decreased by circa 0.6% when comparing data before and after 1994 CE. A predicted decline in hop yield and alpha content of 4–18% and 20–31% by 2050 CE, respectively, calls for immediate adaptation measures to stabilize an ever-growing global sector.

Beer is the world's third most widely consumed beverage after water and tea[1], and traditional beer brewing in central Europe dates back at least to the Neolithic period circa 3500–3100 BC[2]. In addition to water, malting barley and yeast, a much more expensive hop is needed to give beer its incomparable taste[3]. The specific hop aroma emerges from its bitter acid content and many other compounds, including essential oils and polyphenols[4–6]. Changes in alpha bitter acids affect the quality of hops[7–12], and there has been a recent change in consumer preference towards beer aromas and flavors that heavily depend on high-quality hops[13,14]. Amplified by the ongoing craft beer popularity[13], this trend contrasts with previous demands for lower alpha content[14]. The recent craft beer expansion therefore not only triggered new microbreweries but also boosted the demand for aromatic hops globally[15,16]. Although linkages between hop production and climate variation have been reported at local to regional scales[9,10,17–27], relatively little is known

about the possible, direct and indirect, effects of a predicted warmer and drier climate on the yield and alpha content of hops.

Since the cultivation of high-quality aroma hops is restricted to relatively small regions with suitable environmental conditions (Fig. 1), there is a serious risk that much of the production will be affected by individual heat waves or drought extremes that are likely to increase under global climate change[28]. Hop farmers can and have responded to climate change by relocating hop gardens to higher elevations and valley locations with higher water tables, building irrigation systems[10], changing the orientation and spacing of crop rows, and even breeding more resistant varieties[29]. Changing the orientation of crop rows and combining irrigation with water-saving soil management practices have proven to be effective adaptation measures in viticulture[30,31]. It is important that the generative phase of hop plants occurs only in the appropriate photoperiod when sunshine duration is decreasing. This

[1]Global Change Research Institute of the Czech Academy of Sciences, 60300 Brno, Czechia. [2]Czech University of Life Sciences Prague, 16500 Prague, Czechia. [3]Czech Hydrometeorological Institute, 14306 Prague, Czechia. [4]Department of Physical Geography and Geoecology, Faculty of Science, Charles University, 12800 Prague, Czechia. [5]Rothamsted Research Station, Harpenden AL52JQ, UK. [6]Department of Geography, University of Cambridge, Cambridge CB23EN, UK. [7]Swiss Federal Research Institute (WSL), 8903 Birmensdorf, Switzerland. [8]Department of Geography, Faculty of Science, Masaryk University, 61300 Brno, Czechia. ✉e-mail: martin.mozny@chmi.cz

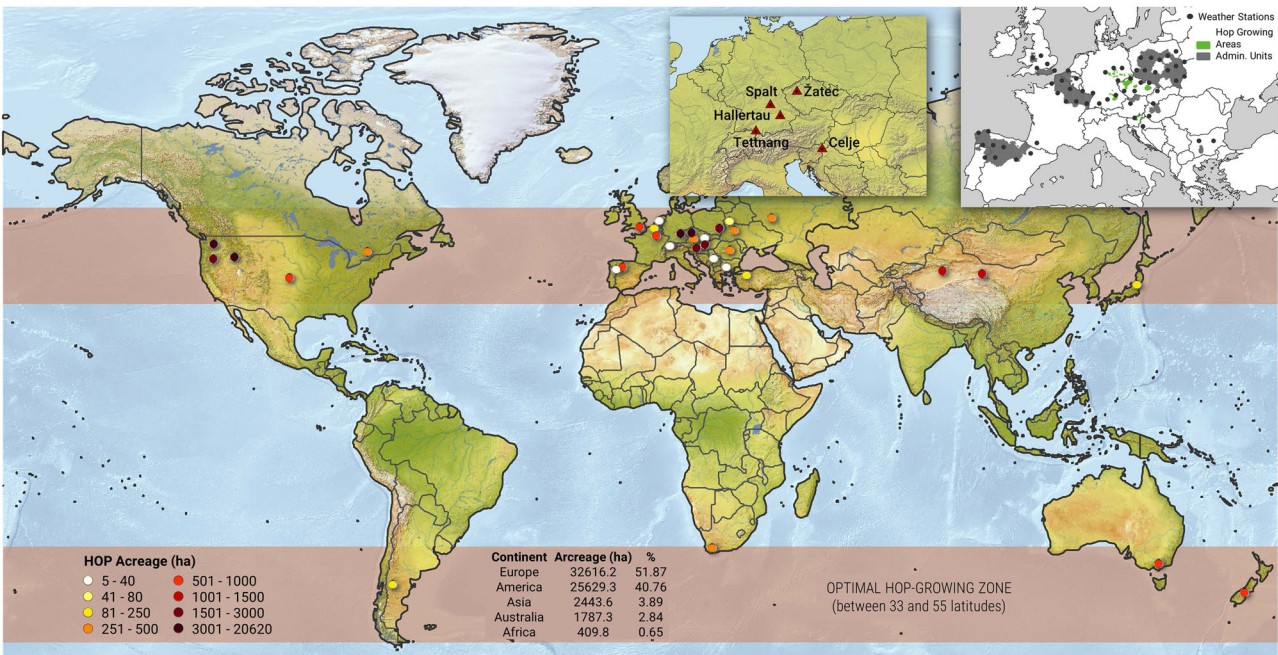

**Fig. 1 | Main map shows the estimated hop acreage in individual countries as well as the approximate position of the optimal hop-growing zone.** The top left map shows five sites representing 3 main German hop-growing regions (Spalt, Hallertau and Tettnang), the Zatec site in Czechia and Celje in Slovenia. The top right map indicates the approximate location of the main hop growing areas in Germany, Czechia and Slovenia as well as recognized hop-producing administrative regions in the remaining countries. The map also includes the locations of the 59 weather stations used in the analysis. The color of the dots responds to the hop acreage and the brown band to the optimal hop-growing zone.

can be achieved by slowing plant growth via growth inhibitors or by building protective shading structures; which is, however, quite expensive. There is a similar problem in vineyards where shading by agrovoltaic panels has been introduced[32,33]. The higher probability of droughts can be partly mitigated by less frequent tillage and cultivation of hop fields, changes in fertilization and the use of row cover crops to support root growth[34]. A systematic and European-wide investigation of the impact of ongoing and predicted climate change on the quality and quantity of aroma hops is, however, still missing.

Here, we show how temperature and precipitation control the yield, alpha content and cone development of aroma hops in Germany, the Czech Republic and Slovenia between 1970 and 2050 CE. We simulate the effect of weather conditions on hop yield and alpha content with a newly developed model. The model requires air temperature and precipitation records for input. Using simulations of future climate, we predict yield and alpha content. We then discuss how hop farmers can implement innovative adaptation measures to stabilize international markets under predicted global warming.

## Results and discussion

Comparison of the average annual yield of European aroma hops during two independent periods, 1971–1994 and 1995–2018, reveals a significant production decrease in the range of 0.13–0.27 t/ha (Fig. 2). The average annual hop yield decreased after 1995 by 19.4% in Celje, 19.1% in Spalt, 13.7% in Hallertau, and 9.5% in Tettnang, whereas the production remained stable in Zatec (0.05 t/ha). In addition to the observed production decline, there were significant declines in the alpha content in all regions between from 0.46 and 1.86% (Fig. 2). The average alpha content decreased by 34.8% in Celje, 15.6% in Hallertau, 15% in Tettnang, 11.5% in Spalt, and 10.5% in Zatec. Hop yields between 1995 and 2018 exceeded the average yields from the 1971–1994 early period only once in Spalt and twice in Celje. Similarly, the average alpha content from the 1971–1994 early period was exceeded only one once in Celje and twice in

Tettnang between 1995 and 2018. Declines in hop yields of more than 30% were recorded in 2000 and 2003. In 2006 and 2015, the decrease in alpha content was greater than 40%. Rising temperatures shifted the onset of the hop growing season by 13 days from 1970 to 2018. After 1995, the average onset of clone development (BBCH 71) occurred earlier compared to that in 1971–1994: 31 days earlier in Celje, 22 days earlier in Zatec, 16 days earlier in Hallertau and Spalt, and 13 days earlier in Tettnang (Fig. 2). These phenological changes shifted the critical ripening period towards the warmer part of the season, which had a negative impact on the alpha content.

To further assess the effects of changing weather conditions on the yields and alpha content of aromatic hops, we developed a parsimonious model simulating the variation in yields and alpha contents based on the difference in the precipitation and temperature from the optimal conditions during the growing season. High yields and alpha contents were obtained in years when weather conditions were close to optimum conditions, whereas low values occurred in years with extreme weather conditions. Hop yields increased with precipitation but decreased after ~15% of normal precipitation, whereas alpha contents decreased linearly with increasing temperatures. We found a statistically significant correlation ($r = 0.41–0.73$; $p < 0.01$) between the difference in the precipitation total from the optimum conditions in the growing season and hop yields (Supplementary Fig. 1). Moreover, a strong correlation was found between the average temperature at the time of heading and the alpha content ($r = 0.61–0.78$; $p < 0.01$; 1979–2018) (Supplementary Fig. 2). We found that sunshine duration at the time of heading was positively correlated with alpha content at $p < 0.01$. The lowest hop yields were negatively affected by a lack of precipitation, while the lowest values of alpha content were caused by extremely high temperatures. High temperatures and sunshine duration caused a sharp drop in alpha content in 2006 in all study areas. The estimated model risk of a decrease in alpha content was the consistently low production that occurred in 2006 in all hop regions in

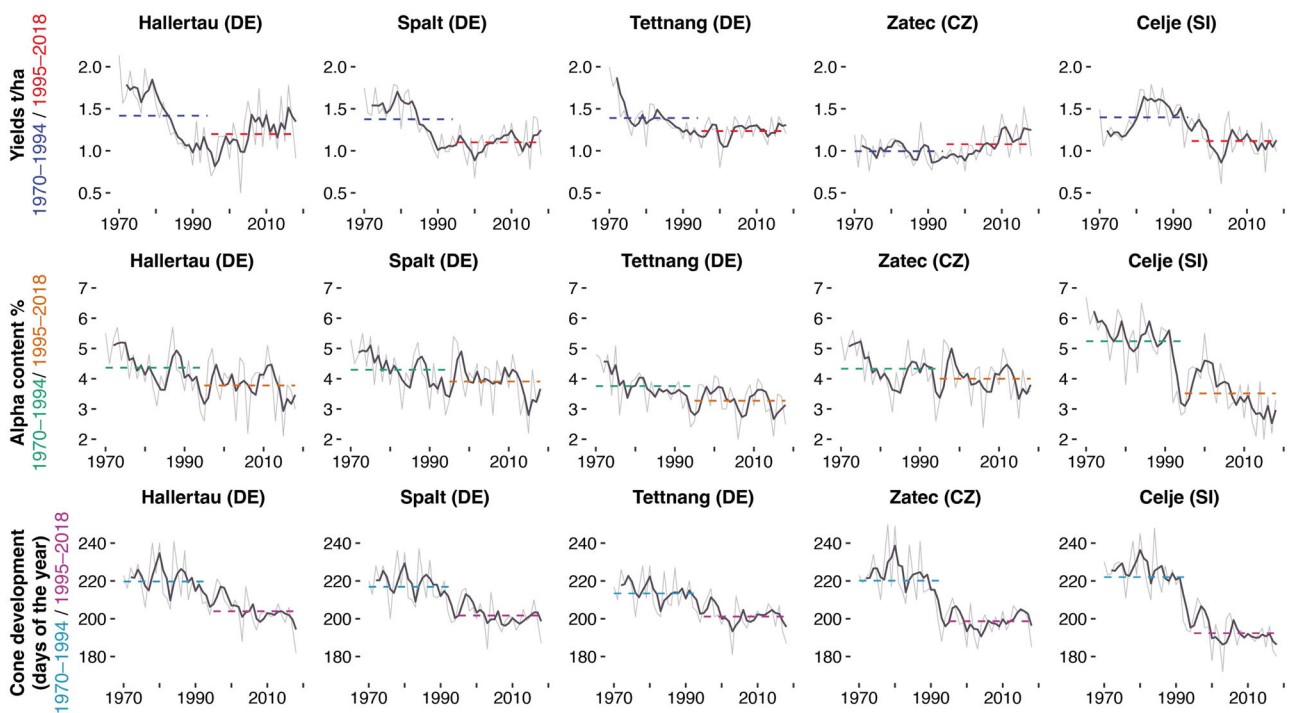

**Fig. 2 | Change in the alpha content [%], date of cone development [days] and yield [t/ha] between 1970 and 2018 in Germany, Czechia and Slovenia.** Average values calculated for two periods (1970-1994 and 1995-2018): yields (dashed blue and red), alpha (dashed green and brown) and cone development (dashed turquoise and purple).

Europe. When temperature and light extremes occurred, most growing areas tended to be negatively affected, and production from other areas could not cover any losses.

Model projections for 2021–2050 suggested a decline in hop yields from 4.1–18.4% when compared to 1989–2018 (Fig. 3). A decrease of 20–30.8% was also projected for alpha content (Fig. 3). The most pronounced declines are expected to occur in the southern hop growing regions in southern Germany and Slovenia (Tettnang and Celje), while the more northern sites in Germany and the Czech Republic (Hallertau, Spalt and Zatec) are expected to experience less pronounced decreases in both parameters. The projected decrease will be caused mainly by rising temperatures and more frequent and severe droughts. These factors will most likely cause a significant decrease in alpha/ha production of 25.3–39.5% compared to current values (Fig. 3).

Changes in hop production and alpha content across Europe and the British Isles support our simulation results (Fig. 4). All scenarios predicted a decline in hop yields between 12% and 35% over 2021–2050 across all major hop growing regions in Europe, with Slovenia, Portugal, and Spain exhibiting the most pronounced declines (Supplementary Fig. 2). The alpha content was predicted to decrease considerably across all regions (Supplementary Fig. 2). Moderate decreases in yield and alpha content were predicted for Germany, the Czech Republic, and Poland, while the strongest declines in hop productivity were predicted for Portugal, Slovenia and Croatia (Supplementary Fig. 2).

Due to changes in climate and water availability for agriculture, some authors have addressed water stress and its influence on crops[28,34–37]. Quantification of the effect of deficit irrigation on hop yields, quality and profitability was performed in Washington State, USA. The results show that plants generally responded to water deficit with yield reductions. The 60% irrigation level caused reductions of 19–33% in the 2-year total yield, while the 80% irrigation level caused total 2-year yield changes from −14% to +2%. However, the quality of the hop cone was not affected. The concentrations of alpha-acids remained similar at all irrigation levels[34].

In addition to the climatic factors described above, there are other external factors that can affect the yield and alpha content of aroma hops. These include the health condition of the hops, epigenetic adaptation and heredity of the hops, irrigation systems, harvest maturity, habitat conditions, and the regulation of hop growth by properly selected agricultural technology and fertilization[24,38–40]. The location of hop fields and suitable soil are very important, especially in the valleys of rivers and smaller streams, where the water table is generally more stable. The importance of drip irrigation combined with advanced irrigation planning technologies, such as the FORHOPS[41] initiative for stabilizing hop yields has been successfully applied in the Zatec region, where irrigated hops have become dominant since 2015. Moreover, wetter and cooler locations have often been used for new hop fields, while open plateau fields have been reduced.

While assessing future climate and environmental impacts on the quality and quantity of aroma hops, the uncertainty associated with model simulations should also be noted. Increased $CO_2$ concentrations could partially compensate for the effects of drought and support yield growth and leaf area index while improving water use efficiency[42]. However, this effect on hops is still under investigation, and we do not yet have enough evidence and knowledge. New findings in hop physiology, such as the beneficial effect of elevated $CO_2$ on the primary metabolism of hop strobilus[43] and the effects of vernalization and dormancy[44] may help in the future to breed hops that are more resistant. Obtaining more detailed measurements and observations directly from hop gardens will allow more detailed seasonal climate analysis in the future and thus improve existing models.

Since agricultural droughts are projected to increase with high confidence in southern Europe and medium confidence in central Europe[45], it will be necessary to expand the area of aroma hops by 20% compared to the current production area to compensate for a future decline in alpha content (and/or hop production). Some uncertainties also need to be considered, such as how hop growers will adopt climate-smart agriculture[46] that seeks to increase sustainable

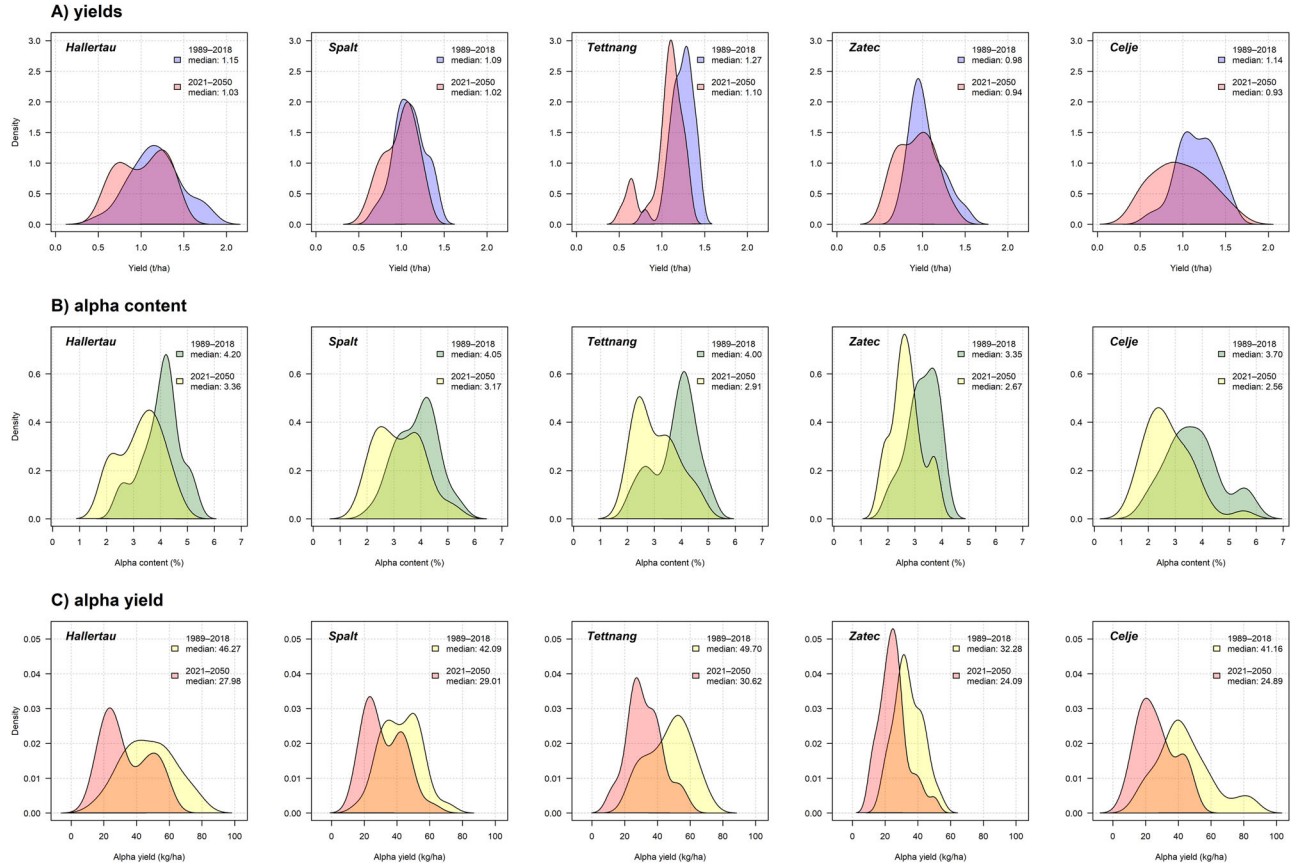

**Fig. 3 | Values of hop yield [t/ha], alpha content [%] and alpha yield [kg/ha] estimated by the models for 1981–2018 (baseline climate) and 2021–2050 (future climate) in Germany, Czechia, and Slovenia.** Median calculated for two periods (1989–2018 and 2021–2050): **A** yields (fill blue and orange), **B** alpha (fill green and yellow) and **C** alpha yields (fill yellow and red).

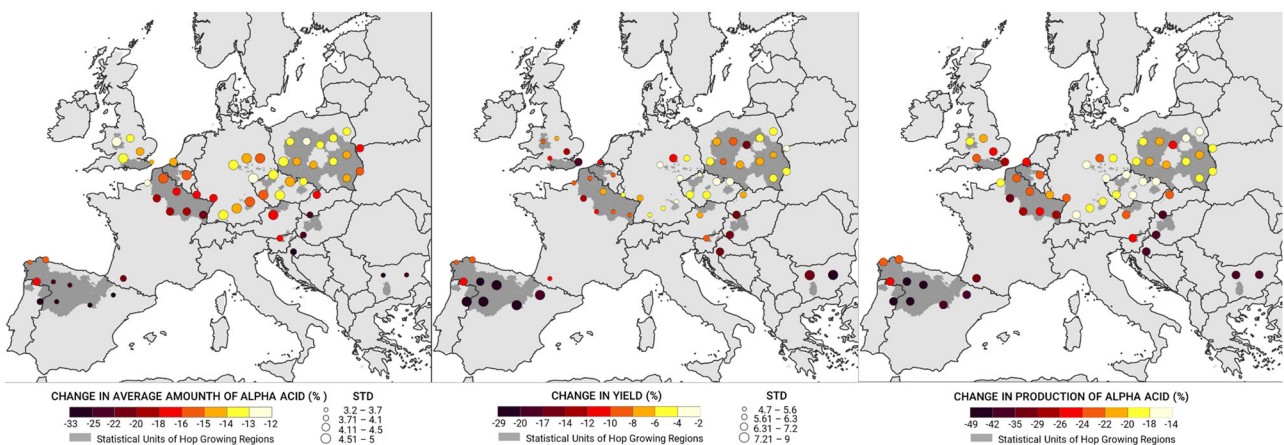

**Fig. 4 | Estimated changes in the amount of alpha acid, hop yield and alpha acid yield (yellow to black gradient) for 59 sites across all hop-producing regions of the EU and UK based on 3 GCM model runs and the RCP 4.5 scenario for** 2021–2050. The size of the color dots responds to the value of the standard deviation (STD).

productivity, strengthen farmer resilience, adapt marketing strategies to customers[47], reduce agricultural greenhouse gas emissions and increase carbon sequestration. All factors will affect the economics and prospects for the future of aroma hop cultivation in Europe.

In summary, this study demonstrates a climate-induced decline in the quality and quantity of traditional aroma hops across Europe and calls for urgent adaptation measures to stabilize international market chains.

## Methods

To identify the current and future impacts of climate change on aromatic hops, we selected important hop-growing areas in Germany (DE), Czechia (CZ) and Slovenia (SLO). These countries cover almost 90% of the total area of aromatic hop fields in Europe. The hop-farming regions of Hallertau (DE), Spalt (DE), Tettnang (DE), Zatec (in German, Saaz; CZ) and Celje (SLO) were selected for this study (Fig. 1). All study areas are located between 46–51°N and 9–15°E in regions with a mild

continental climate. The average annual temperature varies between 8 and 10 °C, and the average annual precipitation fluctuates from 550 to 1050 mm. The altitudes of the hop fields vary between 200 and 500 m ASL, and the predominant soils are clay and loam. Hop acreages, average yields and alpha content (bitter substances in aromatic hops) for the individual areas were obtained from the Barth-Haas reports for 1970–2018[48]. Daily precipitation totals, average daily temperatures, and onset of hop cone development were obtained from the German, Czech and Slovenian national meteorological services. Using regression kriging, which uses altitude as a predictor, we interpolated temperature and precipitation measurements to a resolution of 1*1 km. Daily sunshine duration totals were collected from EUMETSAT satellite observations SARAH 2.1 with a resolution of 0.05 * 0.05°[49]. The obtained layers were then used to calculate the average daily and monthly temperatures and precipitation totals for the selected hop-farming areas.

To assess the effects of weather conditions on the yields and alpha content of aroma hops, we developed a simple model (see SI) simulating the risk of lower yields based on the difference in the rainfall amounts from optimal conditions during the growing season and lower alpha contents based on the temperatures during cone development. Details of the models used are reported in the supplementary information. The formulation of the models is described here in Supplementary Table 1, and the results of the model calibration are shown in Supplementary Fig. 1.

Three global circulation models (GCMs) from the CMIP5 ensemble were used to represent the known variability in the rate of temperature change and precipitation patterns. The model CSIRO-MK36 (CSIRO) was used as the moderate estimate, GISS-E2-R-CC (GISS) represented a lower rate of temperature change and modest increase in dryness, and HadGEM2-ES (HadGEM) represented higher climate sensitivity with a hotter and drier climate. The representative concentration pathway, RCP4.5, was used for the construction of local-scale climate scenarios with the climate models. The climate projections from the GCMs were downscaled to local-scale daily weather by the LARS-WG 6.0 weather generator using the ELPIS dataset of site-specific parameters across Europe[50]. The combined average of the three global mean emissions models (CSIRO MK36; GISS-E2-R-CC; and HadGEM2-ES) was used to simulate the future evolution of yields and alpha contents in 2021–2050 (future) for comparison with those in 1989-2018 (present). The analysis is presented in detail for 5 selected sites and then for 59 ELPIS sites representing all EU and UK hop-producing regions.

## Reporting summary

Further information on research design is available in the Nature Portfolio Reporting Summary linked to this article.

## Data availability

All data that support the findings of this study are publicly available in FigShare (https://doi.org/10.6084/m9.figshare.23180342)[51].

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

## Acknowledgements

This work was supported by the project SustES—Adaptation strategies for sustainable ecosystem services and food security under adverse environmental conditions (CZ.02.1.01/0.0/0.0/16_019/0000797) to M.T., the Technology Agency of the Czech Republic (No. SS02030040, SS02030018) to M.M.

## Author contributions

The authors confirm contribution to the paper as follows: study conception and design: M.M., M.T., and U.B.; computation: M.M., V.V., and T.C.; data collection: V.V., M.S., L.H., and D.S. Analysis and interpretation of results: M.M., M.T., U.B., V.V., and Z.Z. All authors reviewed the results and approved the final version of the manuscript.

## Competing interests

The authors declare no competing interests.
