## [Peer Review File · Nature Communications]

Reviewers' Comments:

Reviewer #1:

Remarks to the Author:

Manuscript#: NCOMMS-23-18831

Title: Climate-induced declines in the quality and quantity of European hops call for immediate adaptation measures

Corresponding Author: Martin Mozny

Key results

Study of the effects of ongoing and predicted climate changes (between 1970 and 2050) on the yields of hop production and on the formation of essential hop constituents for beer flavour has resulted in quantitative assessments of the climate sensitivity on these foremost attributes. The data gathered anticipate a wake-up call for urgent interventions.

Validity

A great number of data has been collected referring to a combination of meteorological measurements, hop contents and model projections. The comprehensible clarifications provide a valid guidance to consider appropriate measures to be applied. The conclusions of the paper call for immediate adaptation in order to stabilize an ever-growing global sector of the cultivation of hops which is a crucial ingredient for the brewing of high-quality beers.

Significance

The worldwide brewery sector remains being confronted with all aspects of manufacturing an industrial product especially from integrating various components into an extremely complex and inherently unstable final result which is beer. There is an ever-increasing and impressive wealth of investigations that add to an escalating affluence of knowledge, however far from being complete. A distinct branch of the world's economy notably the inseparable hop and brewing fields should profit from the results of the present work. One of the most intriguing current issues concerns the impact of global climate changes and exactly this survey contributes firmly to proceed further into the right direction.

Data and methodology

The authors have chosen a correct approach to thoroughly analyse a crowded database. The technically sound account including the figures and supplementary information is well organized and systematic in thought. The section on Methods, lines 281-318, is very short on account of the information collected from accessible sources. This reviewer as an expert in the area is convinced of the proficiency of the participating scientists.

Analytical approach

Insights into variations needed obviously to apply model projections and, although this reviewer is somewhat outside the scope of his expertise, the understanding and the belief in the acceptance of the methodology are true.

Suggested improvements

Within the present scope of this manuscript additional experiments are superfluous. Sufficient material is presented to convince scientists and instructed persons to cope with the problem and to stimulate them to pursue the designated endeavours.

Clarity and context

The text is fluently readable and readily understandable.

References

The manuscript references cite previous literature adequately. An obligate recent reference is missing, namely a report by Bauerle W. L.: *Humulus Lupulus* L. Strobilus Photosynthetic Capacity and Carbon Assimilation, *Plants* 2023, 12, 1816. <https://doi.org/10.3390/plants12091816>; same author as quoted in ref. 38. The study assesses responses to CO₂ and light with respect to hop breeding in response to environmental signals. It was found that the estimated strobilus carbon uptakes increase with elevated carbon budgets under current and future atmospheric CO₂ conditions. A comment by the authors is desirable.

Expertise and constructive feedback

The present reviewer is a globally recognised expert on the chemistry of the hop plant which controls beer flavours and certainly on various health-wise applications of hops. The data afforded in the manuscript regarding the evolution of hops from 1970 mention the so-called alpha content (directly derived from ref. 41), which, in fact, refers to part of a larger whole. Within the list of co-authors, geographical and agricultural departments predominate which in the present context is accountable. Own analytical work is mandatory in future studies. This reviewer decidedly suggest to collaborate with competent laboratories, particularly to focus on the environmental effects about the most essential hop constituents comprising bitter acids, essential oils, and polyphenols. Noteworthy are research groups within Czechia who are leading authorities on detailed hop scientific issues.

Further comments

* Abstract, l. 26: "... 90% of all growing regions, our results from Germany, Czechia and Slovenia..." must be corrected to: "... almost 90% of all

European hop-growing regions, our results from Germany, Czechia and Slovenia...” Remark: Poland as the third biggest hop-growing European country is not included in the study. Slovenia is number 4 in Europe.

* I. 140: see under References about the impact of increased CO2 concentrations, cfr. paper by Bauerle.

* Fig. 2 and Fig. 3: “... three most important hop-producing regions in Europe...” must be corrected to: “... Germany, Czechia and Slovenia...” Remark: Poland is the third hop-growing country in Europe.

Reviewer #2:

Remarks to the Author:

The aim of this article is to estimate the impact of climate change (present and future) on hop growing in Europe. The authors developed an agroclimatic model simulating the impact of rainfall and temperature on hop yields and quality in relation to optimal conditions during the growing season. This model has been applied to a past period and to the future on the basis of outputs from 3 global climate models. Future projections from global climate models have provided different data, particularly on the future intensity of droughts.

The results of this work have demonstrated that climate change has many effects on the phenology, yields and quality of hops. This approach, based on observed and modelled data, is robust and has produced some original and very interesting results. Very few studies have addressed the impact of climate change on hops, and the results are likely to be important for hop growers.

Nevertheless, a few additions seem necessary:

- The authors looked specifically at hops grown in Europe. However, scientific studies have been carried out in other hops producing regions (e.g. USA, New Zealand, etc.) with different impacts of climate change. It would be good to provide a little more information (e.g., more references) on research outside Europe;

- The model was developed specifically on the basis of the difference between precipitation and temperature compared to optimal conditions during the hops growing season. The results relate specifically to the periods of heading, ripening and yield. However, the other stages of hop growth are also strongly affected by climate change, particularly the increase in temperature. For example, in the Nelson region (New Zealand), a major problem for hop growers is the lack of winter cold (dormancy) and the flowering period ((impact on fruit development). Although this article does not address other phenological stages, the authors should perhaps mention this in the discussion, in particular that a seasonal climatic analysis would improve the results.

Please, see "William L. Bauerle. Disentangling photoperiod from hop vernalization and dormancy for global production and speed breeding. Scientific Reports, 2019; 9 (1) DOI: 10.1038/s41598-019-52548-0";

- For future climate projections, the authors used temperature and precipitation data from 3 GCM (CMIP6). All these future projections simulate an increase (from moderate to high) in temperatures, rainfall and drought. Projections of future precipitation between Mediterranean Europe and Northern Europe have significant biases. Additional information on the biases of the CMGs used (perhaps in comparison with CMIP6 Ensemble) would be interesting.

In conclusion, this is a very interesting and original article, but it needs to be improved for publication in the "Nature Communications" Journal.

Manuscript#: NCOMMS-23-18831

Title: Climate-induced declines in the quality and quantity of European hops call for immediate adaptation measures

Corresponding Author: Martin Mozny

Reviewer #1 (Remarks to the Author):

Key results

Study of the effects of ongoing and predicted climate changes (between 1970 and 2050) on the yields of hop production and on the formation of essential hop constituents for beer flavour has resulted in quantitative assessments of the climate sensitivity on these foremost attributes. The data gathered anticipate a wake-up call for urgent interventions.

Validity

A great number of data has been collected referring to a combination of meteorological measurements, hop contents and model projections. The comprehensible clarifications provide a valid guidance to consider appropriate measures to be applied. The conclusions of the paper call for immediate adaptation in order to stabilize an ever-growing global sector of the cultivation of hops which is a crucial ingredient for the brewing of high- quality beers.

Significance

The worldwide brewery sector remains being confronted with all aspects of manufacturing an industrial product especially from integrating various components into an extremely complex and inherently unstable final result which is beer. There is an ever-increasing and impressive wealth of investigations that add to an escalating affluence of knowledge, however far from being complete. A distinct branch of the world's economy notably the inseparable hop and brewing fields should profit from the results of the present work. One of the most intriguing current issues concerns the impact of global climate changes and exactly this survey contributes firmly to proceed further into the right direction.

Data and methodology

The authors have chosen a correct approach to thoroughly analyse a crowded database. The technically sound account including the figures and supplementary information is well organized and systematic in thought. The section on Methods, lines 281-318, is very short on account of the information collected from accessible sources. This reviewer as an expert in the area is convinced of the proficiency of the participating scientists.

Analytical approach

Insights into variations needed obviously to apply model projections and, although this reviewer is somewhat outside the scope of his expertise, the understanding and the belief in the acceptance of the methodology are true.

Suggested improvements

Within the present scope of this manuscript additional experiments are superfluous. Sufficient material is presented to convince scientists and instructed persons to cope with the problem and to stimulate them to pursue the designated endeavours.

Clarity and context

The text is fluently readable and readily understandable.

References

The manuscript references cite previous literature adequately. An obligate recent reference is missing, namely a report by Bauerle W. L.: Humulus Lupulus L. Strobilus Photosynthetic Capacity and Carbon Assimilation, *Plants* 2023, 12, 1816. <https://doi.org/10.3390/plants12091816>; same author as quoted in ref. 38. The study assesses responses to CO₂ and light with respect to hop breeding in response to environmental signals. It was found that the estimated strobilus carbon uptakes increase with elevated carbon budgets under current and future atmospheric CO₂ conditions. A comment by the authors is desirable.

- Reference has been added.
- We added following sentences to Results & Discussion:

While assessing future impacts, the uncertainty associated with future scenarios should also be noted. Increased CO₂ concentrations could partially compensate for the effects of drought and support yield growth and leaf area index while improving water use efficiency³⁸. However, this effect on hops is still under investigation, and we do not yet have enough evidence and knowledge. **New findings in hop physiology, such as the beneficial effect of elevated CO₂ on the primary metabolism of hop strobilus⁴³ and the effects of vernalization and dormancy⁴⁴, may help in the future to breed more resistant hops.**

Expertise and constructive feedback

The present reviewer is a globally recognised expert on the chemistry of the hop plant which controls beer flavours and certainly on various health-wise applications of hops. The data afforded in the manuscript regarding the evolution of hops from 1970 mention the so-called alpha content (directly derived from ref. 41), which, in fact, refers to part of a larger whole. Within the list of co-authors, geographical and agricultural departments predominate which in the present context is accountable. Own analytical work is mandatory in future studies. This reviewer decidedly suggest to collaborate with competent laboratories, particularly to focus on the environmental effects about the most essential hop constituents comprising bitter acids, essential oils, and polyphenols. Noteworthy are research groups within Czechia who are leading authorities on detailed hop scientific issues.

- We agree with this point. In addition to the alpha bitter acids, it would be useful to focus on other substances (essential oils and polyphenols) in the future and to establish cooperation with biochemists.

Further comments:

* Abstract, l. 26: "... 90% of all growing regions, our results from Germany, Czechia and Slovenia..." ... must be corrected to: "... almost 90% of all

- Agreed, corrected.

European hop-growing regions, our results from Germany, Czechia and Slovenia...” Remark: Poland as the third biggest hop-growing European country is not included in the study. Slovenia is number 4 in Europe.

* I. 140: see under References about the impact of increased CO2 concentrations, cfr. paper by Bauerle.

- Agreed, added.

* Fig. 2 and Fig. 3: “... three most important hop-producing regions in Europe...” must be corrected to: “... Germany, Czechia and Slovenia...” Remark: Poland is the third hop-growing country in Europe.

- Agreed, corrected.

Reviewer #2 (Remarks to the Author):

The aim of this article is to estimate the impact of climate change (present and future) on hop growing in Europe. The authors developed an agroclimatic model simulating the impact of rainfall and temperature on hop yields and quality in relation to optimal conditions during the growing season. This model has been applied to a past period and to the future on the basis of outputs from 3 global climate models. Future projections from global climate models have provided different data, particularly on the future intensity of droughts.

The results of this work have demonstrated that climate change has many effects on the phenology, yields and quality of hops. This approach, based on observed and modelled data, is robust and has produced some original and very interesting results. Very few studies have addressed the impact of climate change on hops, and the results are likely to be important for hop growers.

Nevertheless, a few additions seem necessary:

* The authors looked specifically at hops grown in Europe. However, scientific studies have been carried out in other hops producing regions (e.g. USA, New Zealand, etc.) with different impacts of climate change. It would be good to provide a little more information (e.g., more references) on research outside Europe;

- Overall, we have added 6 references to the article:

1. Bauerle, W. L. Disentangling photoperiod from hop vernalization and dormancy for global production and speed breeding. *Sci Rep* 9, 16003 (2019).
2. Hieronymus, S. *For the Love of Hops: The Practical Guide to Aroma, Bitterness, and the Culture of Hops.* (Brewers Publications, a division of the Brewers Association, 2012).
3. Bauerle, W. L. *Humulus lupulus L. Strobilus Photosynthetic Capacity and Carbon Assimilation.* *Plants* 12, 1816 (2023).
4. Comi, M. Other agricultures of scale: Social and environmental insights from Yakima Valley hop growers. *Journal of Rural Studies* 80, 543–552 (2020).
5. Van Simaey, K. R. et al. Potential Determinants of Regional Variation of Three American Aroma Hops Grown in the Willamette Valley, Oregon. *Journal of the American Society of Brewing Chemists* 80, 379–388 (2022).
6. Stark, C. & Gillespie, J. Suitability of New Zealand cropping regions to support hop production. Lincoln University (2021). <https://hapi.co.nz/wp->

<content/uploads/2021/08/Suitability-of-New-Zealand-cropping-regions-to-support-hop-production.pdf>

- Addition to the Results & Discussion section:

Some uncertainties also need to be considered, such as how hop growers will adopt climate-smart agriculture⁴⁶ that seeks to increase sustainable productivity, strengthen farmer resilience, **adapt marketing strategies to customers**⁴⁷, reduce agricultural greenhouse gas emissions and increase carbon sequestration. All of these factors will affect the economics and prospects for the future of aroma hop cultivation in Europe.

* The model was developed specifically on the basis of the difference between precipitation and temperature compared to optimal conditions during the hops growing season. The results relate specifically to the periods of heading, ripening and yield. However, the other stages of hop growth are also strongly affected by climate change, particularly the increase in temperature. For example, in the Nelson region (New Zealand), a major problem for hop growers is the lack of winter cold (dormancy) and the flowering period ((impact on fruit development). Although this article does not address other phenological stages, the authors should perhaps mention this in the discussion, in particular that a seasonal climatic analysis would improve the results.

Please, see “William L. Bauerle. Disentangling photoperiod from hop vernalization and dormancy for global production and speed breeding. Scientific Reports, 2019; 9 (1) DOI: 10.1038/s41598-019-52548-0”.

- Reference added into list of references.
- We also analysed the beginning of flowering (BBCH 61), the trend was very similar to the cone development presented in the paper. Unfortunately, long-term phenological observations in the study areas are not detailed enough and observations of some phases are completely missing.
- We have added the following sentence to the text (Results & Discussion):

Obtaining more detailed measurements and observations directly in the hop gardens will allow more detailed seasonal climate analysis in the future and thus improve the existing models.

* For future climate projections, the authors used temperature and precipitation data from 3 GCM (CMIP6). All these future projections simulate an increase (from moderate to high) in temperatures, rainfall and drought. Projections of future precipitation between Mediterranean Europe and Northern Europe have significant biases. Additional information on the biases of the CMGs used (perhaps in comparison with CMIP6 Ensemble) would be interesting.

- To generate local-scale daily climate scenarios in our study, we apply the commonly used downscaling technique which is based on the LARS-WG weather generator and climatic change factors derived from GCMs. First, 'baseline' distributions of climatic variables in LARS-WG are computed using observed daily weather. These distributions are used to generate 'baseline' climate scenarios. Second, change factors (differences between future and baseline climate projections) derived from GCMs are applied to alter 'baseline' distributions, and then perturbed ('future') distributions are used to generate future climate scenarios. Despite biases in GCM outputs, we can assume that, by analysing climate projections for the baseline and future periods, we could derive change factors in climate, which would be free from bias. This is valid only under the assumption that GCM biases are invariant in time.

- In our study, we selected three GCMs out of 19 GCMs from the CMIP5 ensemble available in LARS-WG, GISS-E2-R-CC [8], CSIRO-MK36 [6] and HadGEM2-ES [9], based on climate sensitivity. Figure below shows climate sensitivity of 19 GCMs from CMIP5 for two regions, Mediterranean Europe (MED) and Northern Europe (NEU). For each GCM, we calculated relative changes in annual precipitation (%) versus absolute changes in mean temperature (C degree) between baseline and end of the century climate (RCP8.5 emission) for NEU and MED regions. Selected three GCMs represent well the range in temperature changes observed in the CMIP5 ensemble from “cool” GISS-E2-R-CC [8] to “hot” HadGEM2-ES [9] with CSIRO-MK36 [6] in the middle.

Reviewers' Comments:

Reviewer #2:

Remarks to the Author:

The authors answered my various questions and added to the paper. This is a very interesting and original paper.